# Oral Administration of Bovine Lactoferrin Modulates the Effects of Chronic Stress on the Immune Response of the Lungs

**DOI:** 10.3390/ijms262010000

**Published:** 2025-10-14

**Authors:** Mariazell Yépez-Ortega, Erick José Zárate-Ayón, Crhistian Axel Gutiérrez-Calvillo, Belen Mendoza-Arroyo, Maritza Velásquez-Torres, Judith Pacheco-Yépez, Diana Rodríguez-Vera, María de los Ángeles Gómez-Román, Uri Axel Garcia-Sanchez, Aldo Arturo Reséndiz-Albor, Ivonne Maciel Arciniega-Martínez

**Affiliations:** 1Laboratorio de Inmunonutrición, Sección de Estudios de Posgrado e Investigación, Escuela Superior de Medicina del Instituto Politécnico Nacional, Plan de San Luis esq. Salvador Díaz Mirón s/n, Ciudad de México 11340, Mexico; myepezo1200@alumno.ipn.mx (M.Y.-O.); cgutierrezc1002@alumno.ipn.mx (C.A.G.-C.); mgomezr1701@alumno.ipn.mx (M.d.l.Á.G.-R.); ugarcias1800@alumno.ipn.mx (U.A.G.-S.); 2Laboratorio de Inmunidad de Mucosas, Sección de Estudios de Posgrado e Investigación, Escuela Superior de Medicina del Instituto Politécnico Nacional, Plan de San Luis esq. Salvador Díaz Mirón s/n, Ciudad de México 11340, Mexico; ezaratea1400@alumno.ipn.mx (E.J.Z.-A.); bmendozaa1200@alumno.ipn.mx (B.M.-A.); 3Sección de Estudios de Posgrado e Investigación, Escuela Superior de Medicina del Instituto Politécnico Nacional, Ciudad de México 11340, Mexico; mvelasquezt1500@alumno.ipn.mx (M.V.-T.); jpachecoy@ipn.mx (J.P.-Y.); dianargzvera@gmail.com (D.R.-V.); 4Departamento de Biología Celular, Centro de Investigación y de Estudios Avanzados del IPN (CINVESTAV), Av. IPN No. 2508 Col. San Pedro Zacatenco, Ciudad de México 07360, Mexico; 5Clínica de Nutrición Especializada Béke, San Sebastián, Texcoco 56130, Mexico

**Keywords:** bovine lactoferrin, chronic stress, lung immunity, immunomodulator, tracheobronchial immunoglobulins

## Abstract

Stress is a predisposing factor for pulmonary diseases; however, its effects on the lungs of healthy individuals have not been fully elucidated. Since bovine lactoferrin (bLf) is a powerful immunomodulator, this study aimed to evaluate whether lactoferrin can modulate the effects of chronic stress on humoral and cellular immunity in the lungs. We performed chronic restraint stress (RS) and oral administration of bLf in a BALB/c model, assessing serum corticosterone, body weight, and various lung immunity parameters, including immunoglobulin concentrations in serum and tracheobronchial lavages (TBLs), secretory IgA (S-IgA) levels in TBLs, IgA-secreting plasma cells, relative expression of pIgR, CD4^+^ lymphocyte Th1 and Th2 populations, and antigen-presenting cell (APC) populations in the lungs. Our results demonstrate that stress increases corticosterone and production of total IgA and IgG, while decreasing levels of IgM and S-IgA, promotes a Th1/Th2 profile imbalance, and decreases APC populations. Interestingly, bLf modulates serum corticosterone levels and stress-induced weight loss, and it also modulates humoral and cellular effects produced by chronic stress. These results demonstrate that bLf should be considered a new therapeutic target for further studies, focusing on prophylactic and co-therapeutic administration to treat and prevent respiratory diseases.

## 1. Introduction

In recent decades, stress has been studied and defined by various authors, highlighting the importance of the nature of the stressor (physical or psychological), its duration (acute or chronic), and the individual’s response in an attempt to maintain homeostasis [1]. One of the first events in response to a stress stimulus is the activation of the hypothalamic–pituitary–adrenal axis, which culminates in the release of catecholamines, neurotransmitters, and hormones, particularly corticosterone [2]. This hormone is secreted into the bloodstream and reaches the entire body, stimulating various organs and systems, including the immune system and mucosal surfaces [3].

The lungs are one of the mucosal organs that are constantly interacting with the environment, and numerous antigens enter the respiratory airways during breathing. Thus, the lower respiratory airways have various immune mechanisms that play a crucial role in their proper functioning. Although the lungs are considered an immunologically privileged site due to the absence of resident microbiota on their surfaces, a limited number of low-molecular-weight particles still gain access. Immune cells are essential for maintaining optimal conditions in the alveoli, the functional and structural units of the lung, thereby facilitating successful gas exchange [4,5]. When these conditions are not optimal for the physiological functioning of the lung, it leads to the development of diseases and infections. In accordance with this, evidence suggests that stress could be a predisposing factor for respiratory pathologies [6], such as infectious diseases [7], by promoting an increase in the number of inflammatory infiltrate cells and prompting defects in cellular remodeling in models of chronic inflammation similar to bronchitis [8,9]. It has also been reported that stress also produces exacerbation of diseases such as asthma [10], Chronic Obstructive Pulmonary Disease (COPD), and allergic rhinitis [11]. Despite the numerous pharmaceutical therapies and strategies available to treat respiratory illnesses today, there is still a need for further research on their effectiveness, monetary costs, side effects and the risk–benefit balance for patients [12]. Consequently, some researchers have focused on the use of adjuvants in the treatment and prevention of respiratory diseases. Thus, the administration of bovine lactoferrin (bLf) has been proposed due to its promising properties, which have been demonstrated in several models [13,14].

Lactoferrin is a glycoprotein found in the secretions of all mammals, exhibiting structural homology across species and direct receptor recognition in various cell types, which enables its heterologous administration; therefore, bovine lactoferrin is among the most extensively used in experimental and clinical trials [15,16,17]. It has been demonstrated that bLf exhibits antioxidant, antiviral, antifungal, and antimicrobial activity, as well as immunomodulatory effects, including a reduction in inflammatory infiltrates in infected tissues. Studies performed on different models and anatomical sites have reported that bLf promotes the production of IgM and IgA antibodies, exhibits anti-inflammatory properties in sepsis processes, and possesses immunomodulatory properties to promote the Th1–Th2 balance [18,19,20,21,22,23].

Despite this, there are no studies focused on elucidating the effect of chronic stress in the respiratory airways of healthy individuals, and little is currently known about the modulation of bLf on the lung response to stress. This study demonstrated that bLf modulates weight loss-induced stress in serum and tracheobronchial lavages, as well as S-IgA concentration and IgA^+^ plasma cell populations in the lungs, and affects Th1 and Th2 CD4^+^ lymphocyte percentages and APC populations, in a corticosterone-dependent manner. This study provides a first overview to elucidate the immunomodulatory effects of lactoferrin on the respiratory immunological response during chronic stress, thereby contributing to the development of a novel therapeutic target for treating and preventing infectious and chronic lung diseases.

## 2. Results

### 2.1. Bovine Lactoferrin Counteracts Weight Loss Induced by Stress

Mice were randomly separated into four experimental groups (Figure 1A). They gained weight during the homing week, before the stress protocol (Figure 1B). Once the stress protocol concluded (Figure 1C), RS lost 0.5 ± 0.24 g (*p* value ≤ 0.01).

### 2.2. Bovine Lactoferrin Modulates Corticosterone

Chronic stress increased serum corticosterone (Figure 2) in the RS group (6869.30 ± 684.94 pg/mL; *p* ≤ 0.001), while in the bLf group, these levels were decreased (561.12 ± 191.36 pg/mL; *p* ≤ 0.05) compared to CTL (2086.64 ± 214.68 pg/mL). Interestingly, in the bLf + RS mice group, corticosterone was increased (4288.41 ± 680.55 pg/mL; *p* ≤ 0.01) at a lower concentration than in the RS group. These results suggest that oral bLf administration before and during chronic stress modulates corticosterone release.

### 2.3. Bovine Lactoferrin Modulates the Effect of Chronic Stress on Immunoglobulin Concentrations

Immunoglobulin concentrations of serum and TBL were modified by chronic stress (Figure 3). Total IgA was increased in serum (Figure 3A; 5.55 ± 0.41 µg/mL, *p* ≤ 0.01) and TBL (Figure 3D; 5.55 ± 0.41 µg/mL, *p* ≤ 0.001) versus the CTL group. Serum IgM (Figure 3B) was diminished in RS (8.32 ± 1.49 µg/mL, *p* ≤ 0.001) and bLf + RS (9.57 ± 0.41 µg/mL, *p* ≤ 0.01), as well as IgM TBL levels of the RS group (Figure 3E; 0.16 ± 0.01 µg/mL, *p* ≤ 0.05), in comparison to the CTL group. IgG was increased in the serum of RS (Figure 3C; 19.83 ± 3.57 µg/mL, *p* ≤ 0.01) and TBL (Figure 3F; 3.58 ± 0.31 µg/mL, *p* ≤ 0.001) versus the CTL group. These results demonstrate the bLf’s immunomodulatory properties by up- or downregulating immunoglobulin concentrations in both serum and TBL during stress.

### 2.4. Bovine Lactoferrin Modulates Stress-Induced Diminished S-IgA Release in Tracheobronchial Lavage and Serum

The analysis of S-IgA TBL concentration (Figure 4A; 0.27 ± 0.04 U/A), and the IgA^+^ plasma cell population (Figure 4B; 3.12% ± 0.25%, *p* ≤ 0.001) was diminished both in the RS group versus CTL. In addition to this, relative expression of pIgR is expressed four times more in the bLf + RS group (Figure 3C; 4.49 ± 0.24, *p* ≤ 0.001) versus the CTL group. These results suggest that chronic stress decreases TBL S-IgA levels, which may be a consequence of the decrease in IgA^+^ plasma cell populations in the lung; however, this decrease does not appear to be related to the relative expression of pIgR. Meanwhile, oral administration of bLf before and during stress modulates S-IgA concentrations and the IgA^+^ plasma cell population, similar to the CTL group, and promotes the overexpression of pIgR.

### 2.5. bLf Promotes Th1 Response and Decreases Th2 Established During Stress

Representative dot plots of CD4^+^ T cell populations are shown in Figure 5A. Analysis of the cytokine profile of lung lymphocytes (Figure 5B) revealed a decrease in the CD4^+^/IL-12^+^ population (1.16% ± 0.06%) and an upregulation of CD4^+^/IL-4^+^ (1.28% ± 0.01%) and CD4^+^/IL-10^+^ (1.34% ± 0.01%) in the RS group. In mice administered with bLf, the CD4^+^/IL-1*β*^+^ population was increased (1.73% ± 0.05%). In the bLf + RS group, CD4^+^/IL-1*β*^+^ (1.67 ± 0.05%) and CD4^+^/IL-12^+^ (1.86 ± 0.06%) percentages were increased; remarkably, CD4^+^/IL-4^+^ (1.18 ± 0.01) and CD4^+^/IL-10^+^ (1.14% ± 0.01%) decreased to levels similar to the CTL group. These results demonstrate that chronic stress upregulates Th2 cytokines, while bLf administration modulates by reestablishing Th1/Th2 balance in the cytokine profile.

### 2.6. bLf Modulates CD64^+^/CD86^+^ Cell Populations Promoted by Stress

Representative dot plots of APC populations are shown in Figure 6A. The analysis revealed a decrease in CD64^+^/CD86^+^ cell populations (1.03% ± 0.01%, *p* ≤ 0.001) in the RS group and an increase in the bLf + RS group (1.42% ± 0.02%) compared to the control group (Figure 6B). These results demonstrate bLf’s capability to modulate APC percentages during chronic stress.

## 3. Discussion

Among many other effects related to stress exposure, metabolic deregulation is one of the most extensively studied. Despite contradictory effects on weight variation, the response appears to be dependent on the duration of exposure to the stimulus. The weight lost in the group subjected to chronic stress aligns with previous reports [24]. Evidence suggests that this may be attributed to hypothalamic–pituitary axis (HPA) activation, which mediates a reduction in leptin levels [25], resulting in lower body fat [26] and diminished energy metabolism [27], thereby disrupting the metabolism of carbohydrates, lipids, and food intake-related hormones [28]. In accordance with our results regarding the bLf + RS group, studies indicate that lactoferrin administration ameliorates weight loss associated with chronic diseases, such as hypertension [29] and influenza [13]. Controversially, it has been reported that lactoferrin administration facilitates weight loss in patients with obesity [30] or predisposed to obesity [31]. However, in our experimental model, we propose that the main mechanism by which bLf prevents body weight loss is related to the modulation of stress-related hormones.

We demonstrated that prophylactic and therapeutic administration of bLf modulates serum corticosterone during chronic restraint stress. As is known, one of the first events in response to stress stimulus is the activation of the HPA, releasing neurotransmitters and hormones such as norepinephrine and corticosterone [2,32], which in turn promote changes in almost every body tissue, including the immune system [3,33] and the lung [7,8,11]. We employed the chronic restraint stress protocol in male mice to obtain more consistent results and accurately evaluate the effect of bLf on serum corticosterone levels. Many authors have reported discrepancies in the levels of stress mediators and cellular responses between male and female mice [34,35,36], and the evidence strongly suggests that the stress response can be modified by hormones associated with the estrous cycle [37]. In this regard, bLf is capable of modulating corticosterone in a manner that depends on the stressor, dose, and timing [38,39]. It has been proposed that bLf is endocytosed by enterocytes and transported to the plasma, passing through the blood–brain barrier to reach the cerebrospinal fluid of the choroidal plexus [40,41,42]. Afterwards, bLf modulates the increase in nitric oxide production by upregulation of all nitric oxide synthase isoforms [43]. Then, the anti-stress effect of opioid µ receptors decreases HPA activity, specifically the release of corticoid-releasing hormone (CRH). Still, it does not modify other mediators, such as adrenocorticotropic hormone (ACTH) [44], epinephrine, and glucagon [45,46]. Several models of chronic stress and lactoferrin administration in rodents yield similar findings to those of this study, providing strong evidence that oral bLf administration modulates serum corticosterone levels in response to stress stimuli [45,47,48]. bLf is also capable of modulating corticosterone release by regulating intestinal microbiota. Studies performed in germ-free mouse models have shown that microbiota downregulate serum corticosterone [49,50,51]. Although the microbicidal effects of lactoferrin have been documented in in vitro and in vivo models [52], numerous reports highlight the prebiotic ability of bLf, which promotes the growth of Bacteroides and Parasutterella [53,54]. These effects seem to be related to lactoferrin dose and iron saturation [54,55]. In this regard, the effect of stress on the intestinal microbiota is not well understood, as differences in reports exist depending on the strain of mice, stress model, and duration of exposure [56]. However, further studies are required to determine the exact mechanisms by which bLf modulates the microbiota and whether this is related to corticosterone release in this specific model of chronic stress.

On the other hand, one of the main determinants of the correct functioning of the lung immunological system is the presence of immunoglobulins in tracheobronchial secretions. There are numerous studies examining how stress promotes immunoglobulin production through HPA activation in healthy individuals; however, most of the research focuses on the intestine [57]. In accordance with this, it has been proposed that stress can have diverse effects on serum immunoglobulins in a duration- and intensity-dependent manner. The first immunoglobulin secreted by plasma cells is IgM. In the respiratory system, IgM is primarily active due to the deficiency of other immunoglobulins, as it is found in low concentrations due to the infrequent presence of IgM-producing cells in the bronchial tree, and its size complicates its transudation from the bloodstream to the bronchial lavage [58]. Nevertheless, it has been reported that IgM concentrations may be increased or decreased depending on the model disease in which they have been evaluated, some of which include fibrosis (barely detectable), pneumonitis, and transplant rejection (increased) [59]. In this study, chronic stress was found to promote a decrease in IgM concentrations; however, this effect was modulated by bLf administration, which may promote proper functioning of the defense against potential pathogens in the lung.

Furthermore, IgA plays a crucial role in lung pathologies such as asthma and COPD, as it can influence tolerance to certain antigens. Additionally, it has been reported that the concentration of IgA in the bronchial secretions of asthmatic patients is increased, while in patients with COPD, it is decreased [60]. Previously, an increase in IgA concentrations in intestinal lavages from the duodenum and ileum has been reported during chronic stress [61]. In contrast, Jarillo-Luna found decreased IgA concentrations in small intestine lavages in a mouse model of chronic stress [62]. Otherwise, both authors reported that the intestine is one of the sites most resistant to stress. In our model, an increase in IgA levels in serum and TBL suggests that chronic stress may be modulating the isotype change in IgA-producing cells in the lung, favoring their secretion and transudation from the serum, likely as a protective mechanism against potential pathogens or particles that could compromise the homeostasis of this mucosa. To clarify the mechanism by which there is such an increase in IgA concentrations, it would be necessary to determine the difference in the concentrations of monomeric IgA (mIgA) and dimeric IgA (dIgA).

Since there are no studies focused on elucidating the role of bLf on immunoglobulin modulation in respiratory secretions, our results demonstrate that bLf does not modify the total IgA, IgG, or IgM in serum nor TBL concentration, in contrast to the increase reported in the distal small intestine [63]. This could be explained by the nature of both mucous tissues. Strikingly, the modulation of these immunoglobulins in the stressed groups is remarkable. The bLf antibody modulation observed in our results is similar to that found in the intestinal lavage of the chronic immobilization model [48]. In the lower respiratory airways, it has only been previously reported that bLf increases IL-17-producing cells and enhances IFN-γ-mediated responses [64], as well as reduces lung consolidation and infiltration in bronchial lavage during influenza virus infection [13]. It is also capable of regulating cytokine genes related to IgA production [65]. Even in human trials, bLf has been reported to contribute to protection against viral infections and modulate respiratory immunity [66]. Still, there is no further information regarding the mechanisms of IgM and IgG secretion. Since IgG levels in serum and TBL showed a similar pattern to IgA concentrations, these results suggest that chronic stress modifies IgG change in isotype and transudation mechanisms, which are also modulated by bLf. Nowadays, we are far from clearly understanding how allostasis mechanisms are formed, and this study provides an initial overview of how chronic stress modifies immunoglobulin levels in a healthy individual, while demonstrating that lactoferrin modulates the effects of stress on the humoral immunity of the lungs.

It has been well established that one of the most important mechanisms for maintaining mucosal homeostasis is the secretion of S-IgA. The deficiency of this immunoglobulin is associated with pathogen colonization, asthma, allergic diseases, and the progression of chronic obstructive respiratory disease [67,68,69]. In concordance with these reports, our results showed decreased S-IgA levels in the TBL and a decrease in the percentages of IgA^+^ plasma cells in the RS group. Interestingly, these parameters in the bLf + RS group are similar to those of the control, demonstrating the immunomodulatory effect of bLf. Since S-IgA production and secretion depend on the expression of the polymeric immunoglobulin receptor (pIgR) in lung epithelial cells, we evaluated the expression of the pIgR gene in these cells. pIgR is related to the process of transcytosis and the release of the secretory component (SC) to form S-IgA complex [70]. Several studies have demonstrated that pIgR expression is regulated by some cytokines, such as TNF-α, IFN-γ, TGF-β, as well as signal pathways activated by glucocorticoid receptors [71]. Hence, it has been shown that IL-4 is capable of downregulating bronchial epithelial pIgR expression in asthma [72]. Therefore, the relative expression of the pIgR gene had no difference in the stress group, but it was overexpressed in the bLf + RS group. It has been reported that in an acute stress model, bLf promoted the upregulation of pIgR mRNA, while diminishing protein expression in the proximal intestine. Conversely, in the distal small intestine, pIgR mRNA was not modified, but protein expression increased [47]. Despite this, studies on cultures of primary human bronchial epithelial cells from smokers demonstrated that pIgR protein expression is not necessarily related to its gene expression, and this could be related to post-transcriptional mechanisms [73,74].

In previous studies, the bLf administration modulates IgA^+^ plasma cell populations of the distal small intestine [63]. This is the first report to focus on the effect of stress on lung immunoglobulins, demonstrating bLf’s capability of modulating the decrease in IgA^+^ plasma cell population and S-IgA low expression induced by stress. Nevertheless, further studies are required to clarify if it is related to pIgR protein expression and the mechanisms and specific signaling pathways implicated. These findings suggest that lactoferrin may serve as another protective mechanism, acting as a prophylactic and adjuvant therapeutic agent in lower respiratory tract diseases.

Numerous studies have investigated the role of stress in pulmonary diseases. It has been widely reported that it induces airway inflammation [9] via HPA activation [10]. Consequently, asthma symptoms are exacerbated by the promotion of an increase in CD4^+^ Th2 cells [75] and the further secretion of cytokines, such as IL-4, IL-5, and IL-13 [76,77,78,79]. At the same time, Th1 cytokines are diminished [80,81]. Asthma symptoms improve when a glucocorticoid receptor antagonist is administered [82]. There are also findings in infectious diseases. In a model of the influenza virus, which demonstrates that stress promotes a Th2 profile, thereby decreasing antiviral defense and worsening the disease’s development [83]. Lafuse et al. have reported that stress improves IL-10 levels by inducing psychological stress in mice infected with *Mycobacterium tuberculosis* (MTB), which promotes pathogenicity [84]. We found that T CD4^+^/IL-4^+^ and T CD4^+^/IL-10^+^ populations were highly increased in the RS group, and our results are consistent with these reports, suggesting that chronic stress might be promoting lung susceptibility to infectious and respiratory diseases.

On the other hand, numerous studies have demonstrated the immunomodulatory effects of lactoferrin in various respiratory pathologies by promoting or downregulating inflammatory responses [85]. It has been reported that oral administration of bLf diminished viral load in BALB/c mice infected with the influenza virus, and it also promotes tissue repair response by decreasing cell infiltration [13]. Furthermore, some findings suggest bLf is capable of downregulating asthma symptoms and ovalbumen-induced lung inflammation [86]. In an MTB infection, bLf demonstrated its ability to increase the number of CD4^+^ Th1 cytokine-producing cells in the lung, thereby promoting the amelioration of pathological response and preventing the formation of granuloma [64,87]. Some authors have hypothesized that bLf could be a novel prophylactic and therapeutic agent for ameliorating COVID-19 and other infectious lung diseases [88]. Our results demonstrated that bLf modulates the increase in Th2 lymphocyte populations during chronic stress and promotes a Th1 profile, thereby balancing the cytokine profile in the lung. These effects may help prevent the exacerbation of chronic pulmonary or allergic diseases and reduce the risk of infectious diseases of the lower respiratory airways; however, further studies are required to clearly understand the mechanisms involved in signaling these immunological events.

The role of macrophages and antigen-presenting cells is a notable finding in recent studies aimed at elucidating the mechanisms of pulmonary homeostasis management. Among other functions, these cells are the first to protect the lower respiratory tract against pathogens and coordinate and regulate respiratory secretions [89]. Their presence is also associated with the prevention of asthma exacerbation [90]. There is evidence that stress diminishes APC population, such as dendritic cells [82], and this might be related to a decrease in IL-12 release in response to glucocorticoids [91]. In this study, chronic stress diminished CD64^+^/CD86^+^ cell populations; however, this effect was mitigated by the prophylactic administration of bLf.

The evidence provided in this study demonstrated that oral administration of bLf modulates the effect of chronic stress in the serum and TBL immunoglobulins, prevents the decrease in S-IgA and IgA^+^ plasma cell populations promoted by stress, and counterbalances Th1-Th2 lymphocyte profiles. Even when more studies are conducted to clarify these findings, such as the protein expression of pIgR and the role of macrophages in lung immunity, we hypothesize that this may be a protective effect of bLf, maintaining the innate immune system in a state that allows it to mount a response. This assertion is supported by the fact that bLf administration had no changes without exposure to stress stimulus. Thus, this study provides a first overview of the potential prophylactic and co-therapeutic administration of bLf to prevent low respiratory infectious diseases and amelioration of chronic lung diseases.

## 4. Materials and Methods

### 4.1. Animals

A total of 40 male BALB/c mice (aged 10–12 weeks old and weighing 25–30 g) were used. Mice were housed in transparent polycarbonate boxes with sterile shavings bed and kept on a 12 h light/dark cycle (lights on at 6:00 a.m.) at room temperature at 20 °C, with relative humidity of 55% and provided with water and Purina Lab Diet 5001. Animals were handled according to a protocol (ESM-CICUAL-03/06-09-2020) in accordance with the Mexican federal regulations for animal experimentation and care (NOM-062-ZOO-1999, Ministry of Agriculture, SAGARPA, Mexico City, Mexico), and the experiments were approved by the Institutional Animal Care and Use Committee of the Escuela Superior de Medicina, Instituto Politécnico Nacional.

### 4.2. Stress and bLf Administration Protocol

Mice were randomly divided into four experimental groups with n = 10: (a) Control (CTL): kept for 14 days in housing with water and food ad libitum, but this group was devoid of drinking and food intake while RS was performed. (b) Restraint stress (RS): mice were maintained for six more days with minimal manipulation, and then, they were introduced into 9 cm large, 3 cm high, and 3.5 cm diameter cylindrical plexiglass movement restriction chambers with many holes for adequate ventilation [62] for four hours, from 8:00 a.m. to 12:00 p.m., for eight consecutive days [92]. (c) bLf: A total of 5 mg of bovine lactoferrin diluted in 100 µL of vehicle (sterile water) was administered by buccal deposition daily, for a period of 14 consecutive days [63,93]. (d) bLf + RS: A total of 5 mg of bovine lactoferrin diluted in 100 µL of sterile water was administered daily, for a period of six days before stress induction. Administration of bLf continued for eight more days (for a total of 14 days), during which the mice were subjected to movement restriction stress for four hours (Figure 1A).

### 4.3. Mice Weight

Mice were weighed first upon arrival at the laboratory, before the stress protocol, and after the experimental model, right before euthanasia. The weight registered at the beginning of each week was subtracted from the next measure to analyze weight gain or loss (Figure 1B,C).

### 4.4. Blood Collection

Each mouse was euthanized by an intraperitoneal injection of a lethal dose of 100 mg/kg body weight pentobarbital sodium salt (cat. P3761, Sigma-Aldrich, Darmstadt, Germany) and exsanguinated by cardiac puncture. Approximately 1 mL of blood was collected and centrifuged at 3000 rpm for 10 min to obtain the serum, and it was stored at −20 °C for further use.

### 4.5. Corticosterone Assay

The corticosterone concentration in the plasma was determined using a commercially available ELISA kit according to the manufacturer’s instructions (cat. no. 501320 Cayman Chemical, Ann Arbor, MI, USA). Plasma samples’ corticosterone concentrations were calculated based on a standard curve and were expressed in pg/mL (Figure 2).

### 4.6. Tracheobronchial Lavage

Once exsanguination was performed, the mouse was fixed on a dissection table, and the thoracic cavity was exposed. An incision was made in the trachea by inserting a metal cannula, which was used to introduce 1 mL of 1X PBS into the lungs. Gentle massage and aspiration were then performed, and the lavage was recovered, placed in a microtube, and stored. The lavage was centrifuged at 1500 rpm at 4 °C for 10 min. The supernatant was collected and stored at −20 °C with brief modification [94].

### 4.7. Immunoglobulin Measurement

Concentrations of total IgM, IgG, and IgA, as well as secretory IgA (S-IgA), were determined in the tracheobronchial lavages by performing a sandwich ELISA with brief modifications [63]. The plate was coated with the capture antibody (anti-IgA, anti-IgM, anti-IgG) and incubated. HRP-coupled antibodies were added: anti-IgA (HRP goat anti-mouse cat. 626720 Life technologies, Carlsbad, CA, USA), anti-IgM (HRP goat anti-mouse cat. M31507 Life technologies, Carlsbad, CA, USA), anti-IgG (HRP goat anti-mouse cat. 626520), and anti-secretory component (HRP goat anti-mouse cat. sc-374343, Santacruz, Dallas, TX, USA). Absorbance was measured at 490 nm using an enzyme-linked immunosorbent assay reader (Sigma).

### 4.8. Lung Lymphocyte Cells Purification

Lungs were removed and incubated in 15 mL of 1X RPMI 1640 (cat. 23400-062, GIBCO, Carlsbad, CA, USA) supplemented with 1% fetal bovine serum (FBS), with 100 µM ethylenediaminetetraacetic acid (8993-01 J.T. Baker, Phillipsburg, NJ, USA) and 1 mM dithiothreitol (D9779, Sigma-Aldrich, Darmstadt, Germany) for 30 min at 37 °C. After incubation, tissue dissociation was performed using a plunger and a steel mesh, followed by filtration and centrifugation. The cell button was subjected to 75%/40% Percoll gradients to obtain leukocytes and 40%/20% Percoll gradients to obtain epithelial cells. Interphase cell rings were collected and washed to obtain a pellet [95].

### 4.9. Flow Cytometry Assay

Suspension of 1 × 10^6^ cells was incubated with corresponding extracellular markers for plasma cells (anti-CD19/PE cat. 553786, anti-CD138/APC cat. 558626 BD Biosciences, Franklin Lakes, NJ, USA), APC (anti-CD64/PE cat. 558455, BD Biosciences, anti-CD86 cat. 105012, BioLegend, San Diego, CA, USA), and T CD4^+^ cells (anti-CD4/PerCP cat. 100538, BioLegend). Subsequently, APCs were fixed with 4% paraformaldehyde. T lymphocytes and plasma cells were fixed and permeabilized by incubating for 20 min in the dark with Cytofix/Cytoperm (cat. 554722, BD Biosciences) and centrifuged at 1500 rpm at 4 °C for 5 min. Then, cells were washed with Perm Wash (cat. 554723, BD Biosciences) and incubated with antibody cocktails, respectively: anti-IgA/FITC (Cat. 559354, BD Biosciences) for plasma cells and Th1 (anti-IL-1*β*/FITC cat. IC413F, R&D Systems, anti-IL-12/APC cat. 554480, BD Biosciences) and Th2 (anti-IL-4/PE cat. 554435, BD Biosciences, anti-IL-10/FITC cat. 505005, BioLegend) for each analysis of intracellular cytokines. After incubation, the cell pellet was washed with Perm Wash, and then the samples were fixed and filtered [96]. Samples were stored at 4 °C in the dark until analysis on the FACS ARIA flow cytometer (Beckton Dickinson Company, Franklin Lakes, NJ, USA) with BD FACSDIVA™ v6.1 software (BD Biosciences), acquiring 20,000 gated events from each sample. The data were analyzed using FlowJo v10.10.0 (BD Life Sciences, Franklin Lakes, NJ, USA). Cell percentages were reported as the mean ± SD.

### 4.10. Real-Time qPCR of pIgR

#### 4.10.1. RNA Extraction

The extraction of RNA from leukocytes was carried out by a gradient of TRIzol reagent (cat. Invitrogen™, 15596026, Life Technologies, Carlsbad, CA, USA) and the chloroform technique [97].

#### 4.10.2. cDNA Synthesis

RNA was treated with the RQ1 RNase-Free Dnase Kit (cat. M6101, ThermoScientific, Carlsbad, CA, USA) following the manufacturer’s instructions. The synthesis of the cDNA was made using a commercial kit (RevertAid First Strand cDNA Synthesis kit, cat. K1622, ThermoScientific) by following the manufacturer’s instructions. cDNA samples were stored at −70 °C.

#### 4.10.3. Quantitative Real-Time Polymerase Chain Reaction (qRT-PCR)

The primers for pIgR and GAPDH were designed with Primer Express v.3.0.1 (Applied Biosystems) software and synthesized by UNIPARTS S.A. of C.V. (Table 1). For the PCR reactions, each well was filled with 10 µL of SYBR™ Green PCR Master Mix (cat. 4309155, Applied Biosystems™, Thermo Fischer Scientific, Waltham, MA, USA); 0.2 µL Primer F; 0.2 µL Primer R; 7.6 µL of Water; and 2 µL of the sample. The amplification was carried out in the Step One Real-Time PCR System (Applied Biosystems™, Waltham, MA, USA) and was analyzed with the help of StepOne™ v2.3 software. Data normalization was carried out using the constitutive gene GAPDH for determination.

### 4.11. Statistical Analysis

The results were analyzed using a one-way ANOVA and Tukey’s multiple comparisons test with GraphPad Prism v. 8.0.2. Data are presented as the mean ± standard deviation (SD). Significant differences were defined as *p* ≤ 0.05.

## Figures and Tables

**Figure 1 ijms-26-10000-f001:**
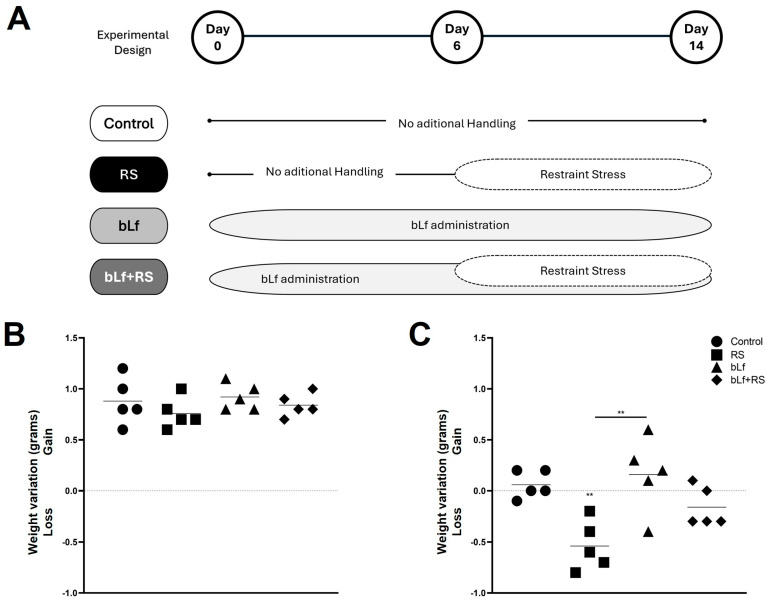
Experimental design and weight monitoring. Mice were divided into four experimental groups: control (CTL), restraint stress (RS), bLf (bLf administered orally), and bLf + RS. (**A**) Graphics represent the mean and ±SD of weight variation from arrival to before the experimental model (**B**) and during the experimental model’s performance (**C**). Mice groups showed no difference in weight gain or loss during the first week; however, after the stress protocol, the RS group lost approximately 0.5 g (** *p* ≤ 0.01).

**Figure 2 ijms-26-10000-f002:**
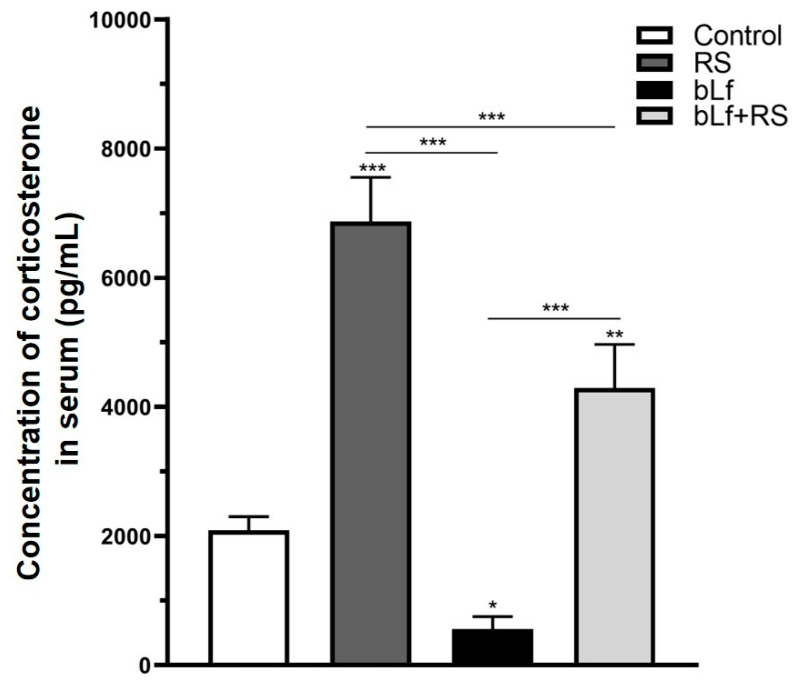
Effect of stress and bLf administration on serum corticosterone. Corticosterone levels were increased in the RS group, which were almost four times those of baseline (*** *p* ≤ 0.001), but in the bLf + RS group, levels were barely doubled (** *p* ≤ 0.01). bLf diminished corticosterone levels by half of those of the control group (* *p* ≤ 0.05).

**Figure 3 ijms-26-10000-f003:**
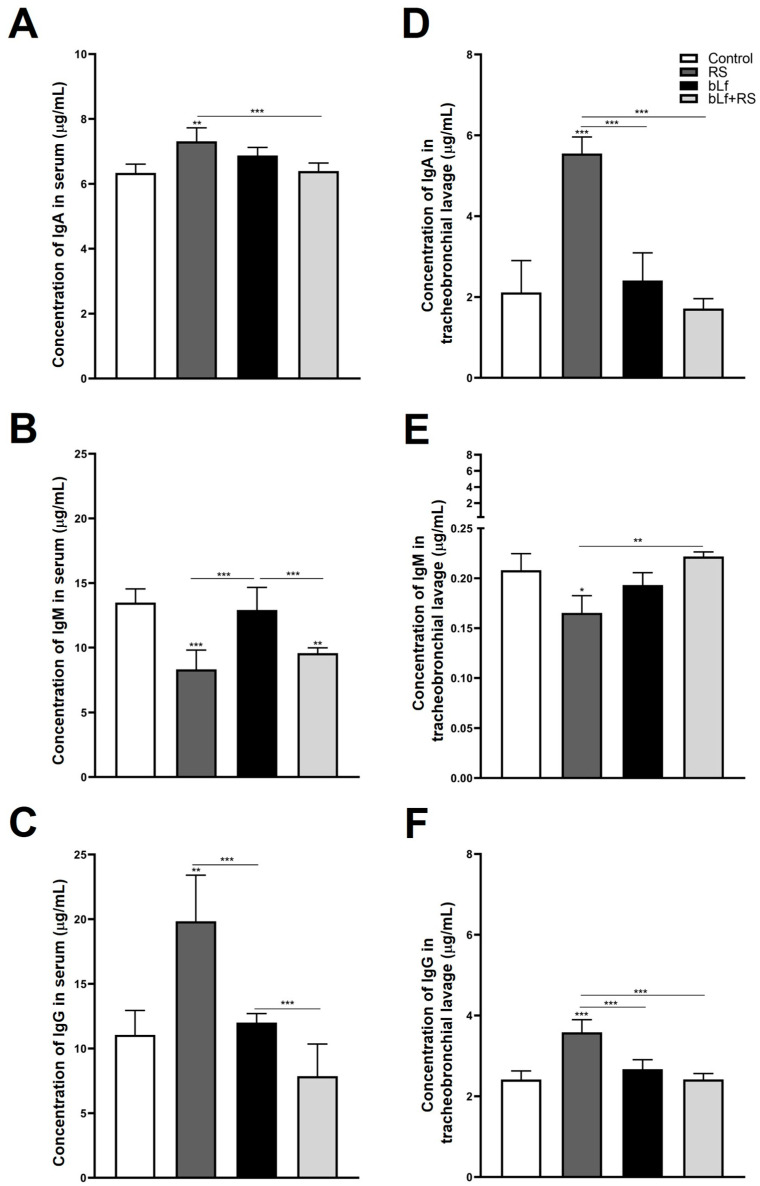
Levels of immunoglobulins in serum and TBL. Total IgA was increased in serum (**A**) and TBL (**D**) of the RS group; total IgM was diminished in serum (**B**) and TBL (**E**) of the RS group; and IgG concentrations were increased in serum (**C**) and TBL (**F**) of the RS group. These modifications in immunoglobulin levels of stressed mice were modulated by the administration of bLf (bLf + RS groups). (* *p* ≤ 0.05; ** *p* ≤ 0.01; *** *p* ≤ 0.001).

**Figure 4 ijms-26-10000-f004:**
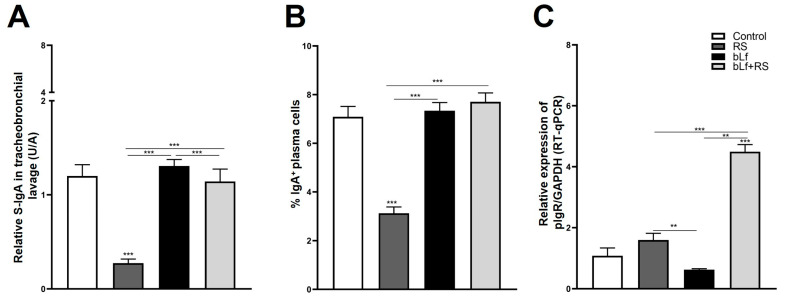
Analysis of S-IgA levels in TBL, secretion, and transport-related mechanisms. RS decreased S-IgA in TBL (**A**) and the percentage of IgA^+^ plasma cell populations (*** *p* ≤ 0.001) (**B**), but these levels are not modified in bLf nor bLf + RS groups. Relative expression of pIgR was increased in bLf + RS (*** *p* ≤ 0.001), and it was decreased in bLf group only compared to RS and bLf+RS groups (** *p* ≤ 0.01) (**C**).

**Figure 5 ijms-26-10000-f005:**
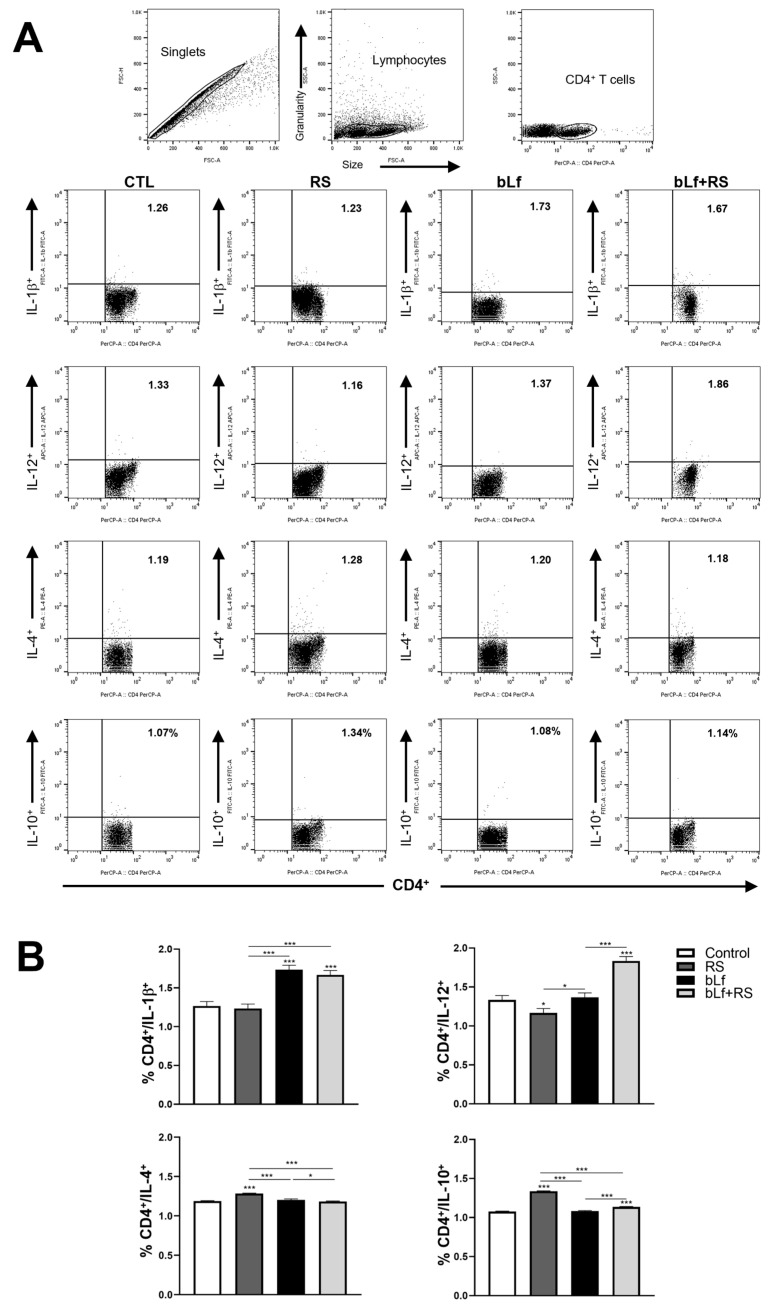
Effect of stress and bLf administration on Th1 and Th2 populations in the lungs. Representative dot plots of CD4^+^ cell populations are shown in (**A**). In the RS group, the CD4^+^/IL-12^+^ population decreased, while CD4^+^/IL-4^+^ and CD4^+^/IL-10^+^ cell percentages increased. bLf administration upregulated CD4^+^/IL-1*β*^+^ populations. In the bLf + RS group, CD4^+^/IL-1*β*^+^ and CD4^+^/IL-12^+^ percentages were increased, but CD4^+^/IL-4^+^ and CD4^+^/IL-10^+^ cell populations decreased (**B**) (* *p* ≤ 0.05, *** *p* ≤ 0.001).

**Figure 6 ijms-26-10000-f006:**
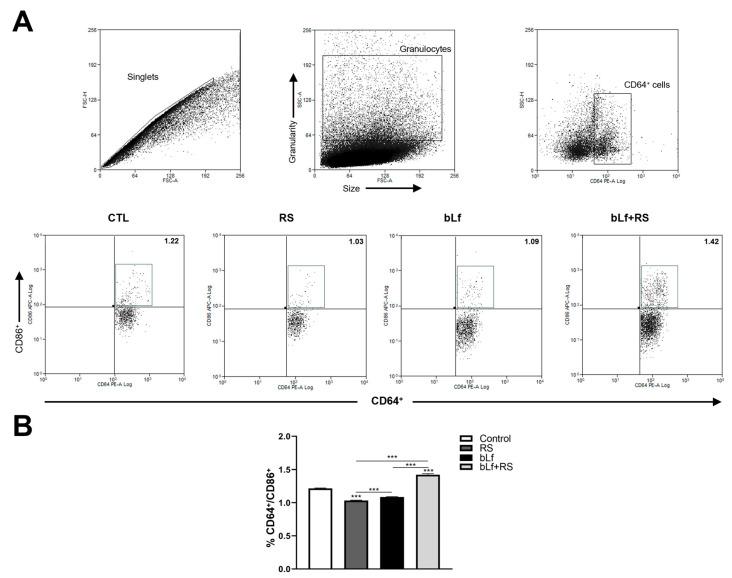
Effect of stress and bLf administration in APC populations in the lungs. (**A**) Representative dot plots of CD64^+^/CD86^+^ cells. (**B**) The percentage of APC decreased in RS (*** *p* ≤ 0.001), but this population increased in the bLf + RS group compared to the control group (*** *p* ≤ 0.001).

**Table 1 ijms-26-10000-t001:** Primers sequences for pIgR and GAPDH used in the RT-qPCR assay.

Primer	Forward 5′-3′	Reverse 3′-5′
pIgR	TCAATCAGCAGCTACAGGACAGA	GTGCACTCCGTGGTAGTCA
GAPDH	GATGCCCCCATGTTTGTGAT	GGTCATGAGCCCTTCCACAAT

## Data Availability

The data are contained within the article.

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
