# Peer review of "Oral Administration of Bovine Lactoferrin Modulates the Effects of Chronic Stress on the Immune Response of the Lungs"

_ijms, 2025, doi:10.3390/ijms262010000_

Round 1

Reviewer 1 Report

Comments and Suggestions for Authors

This study systematically investigates the immunomodulatory potential of bovine lactoferrin (bLf) in counteracting chronic restraint stress-induced pulmonary immunopathology. The authors demonstrate that bLf administration not only alleviates systemic stress markers but also restores multiple aspects of lung immunity

1.The introduction is currently presented as a single, lengthy paragraph. To significantly improve readability and logical flow, I strongly recommend restructuring it into three distinct thematic paragraphs.

2.Please note that the numerical values on the y-axis labels in Figures 3D-F are partially obscured and should be corrected for clarity.

3.It would be valuable to include a statistical analysis comparing the RS group and the bLf+RS group in Figure 3B. Please indicate on the graph whether their difference is statistically significant.

4.The axis label text in Figure 5A is too small and should be enlarged to improve clarity.

5.The finding that bLf + RS significantly upregulates pIgR (polymeric immunoglobulin receptor) expression is particularly noteworthy. To elucidate the underlying mechanisms, it is recommended that further studies in cell models investigate the specific signaling pathways (e.g., TLR4, NF-κB) through which bLf + RS promotes pIgR expression, especially given the contrasting observation that bLf alone downregulates pIgR expression.

Author Response

As requested, all changes made according to the reviewers’ comments and our own corrections were highlighted in red in this revised version.

This study systematically investigates the immunomodulatory potential of bovine lactoferrin (bLf) in counteracting chronic restraint stress-induced pulmonary immunopathology. The authors demonstrate that bLf administration not only alleviates systemic stress markers but also restores multiple aspects of lung immunity

The entire manuscript has been thoroughly revised and edited by the professional Language and Figure Editing service at MDPI Author Services.

  1. The introduction is currently presented as a single, lengthy paragraph. To significantly improve readability and logical flow, I strongly recommend restructuring it into three distinct thematic paragraphs.

Appreciated reviewer, thanks for your comments. We fully agree that dividing the introduction into thematic sections enhances clarity and improves the logical flow. Accordingly, we have restructured it.

2.Please note that the numerical values on the y-axis labels in Figures 3D-F are partially obscured and should be corrected for clarity.

As suggested by the reviewer, we have corrected the formatting of Figures 3D–F to ensure that all numerical values on the y-axis are now clearly visible and legible.

3.It would be valuable to include a statistical analysis comparing the RS group and the bLf+RS group in Figure 3B. Please indicate on the graph whether their difference is statistically significant.

As indicated by the reviewer, we have carried out an ANOVA and Turkey’s multiple comparisons test (between every group). There is no statistically significant difference between RS and bLf+RS groups (p=0.423).

4.The axis label text in Figure 5A is too small and should be enlarged to improve clarity.

As suggested by the reviewer, we have enlarged the axis label text in Figure 5A to ensure better readability and consistency with the other figures.

5.The finding that bLf + RS significantly upregulates pIgR (polymeric immunoglobulin receptor) expression is particularly noteworthy. To elucidate the underlying mechanisms, it is recommended that further studies in cell models investigate the specific signaling pathways (e.g., TLR4, NF-κB) through which bLf + RS promotes pIgR expression, especially given the contrasting observation that bLf alone downregulates pIgR expression.

We are grateful to the reviewer for highlighting this important observation and for suggesting future directions to explore the underlying mechanisms. Previously in the manuscript, we mentioned in the discussion section thatSeveral studies have demonstrated that pIgR expression is regulated by some cytokines, such as TNF-α, IFN-γ, TGF-β, as well as signal pathways activated by glucocorticoid receptors. Hence, it has been shown that IL-4 is capable of downregulating bronchial epithelial pIgR expression in asthma”.

Therefore, the relative expression of the pIgR gene had no difference in the stress group, but it was overexpressed in the bLf+RS group. It has been reported that in an acute stress model, bLf promoted the upregulation of pIgR mRNA, while diminishing protein expression in the proximal intestine. Conversely, in the distal small intestine, pIgR mRNA was not modified, but protein expression increased. Despite this, studies on cultures of primary human bronchial epithelial cells from smokers demonstrated that pIgR protein expression is not necessarily related to its gene expression, and this could be related to post-transcriptional mechanisms”

We also mentioned that one of the limitations of this work is that “Nevertheless, further studies are required to clarify if it is related to pIgR protein expression and the mechanisms and specific signaling pathways implicated”. We agree with your observations about the need to study specific pathway mechanisms related to pIgR expression, and we have added it to the text.

In addition, with this we have calculated statistical analysis (ANOVA and Turkey’s multiple comparisons). The comparison between CTL and bLf groups had no statistically significant difference (p=0.1010). Despite this, it will be interesting to study whether the gene and protein expression are related in our model.

Reviewer 2 Report

Comments and Suggestions for Authors

The manuscript  “Oral administration of bovine lactoferrin modulates the effects of chronic stress on the lung immune response” by Mariazell Yépez-Ortega  et al. adds new information on immune reconstituting properties of lactoferrin. Although the work deals predominantly with the effects of Lf on the status of the innate immunity of lungs, modified by the immobilization stress,  it  also measures levels of  serum Ig and corticosterone and body weight. So, the  systemic parameters are also determined.

The language needs correction by a native speaker. There are numerous grammar and stylish errors e.g. “The analysis of CD 64/86 cell populations (Fig. 6B) diminished in RS, while it was increased in bLF+RS “.  The manuscript is difficult to read. The Introduction should be divided by several sections.

No sex of the mice is given. Males and females respond differently to stress (Yajun Qiao, et al. A Study of Sex Differences in the Biological Pathways of Stress Regulation in Mice CNS Neurosci Ther. 2025;31(5):e70433)

What do you mean by “oral deposition” . May be “buccal”?

In the control group the mice should be also devoid of drinking and eating during the “stress periods”

More details is needed regarding the immobilization cages. 50 ml conical, perforated tubes?

section 2.3. Fig 3B does not show results from TBL but from serum

The discussion does not mention potential advantageous effects of LF on the intestinal microbiota  (Griffiths EA,  et al. In vitro growth responses of bifidobacteria and enteropathogens to bovine and human lactoferrin. Dig Dis Sci 48: 1324-1332, 2003.; Kawasaki Y, et al.. Inhibitory effects of bovine lactoferrin on the adherence of enterotoxigenic Escherichia coli to host cells. Biosci Biotechnol Biochem 64: 348-354, 2000.)  which may have a positive effect on the intestine microbiota during stress (Karla Vagnerová  et al. Interactions Between Gut Microbiota and Acute Restraint Stress in Peripheral Structures of the Hypothalamic-Pituitary-Adrenal Axis and the Intestine of Male Mice.  Front Immunol. 2019, 19:10:2655.) and consequently ameliorate negative effects of stress.

Author Response

As requested, all changes made according to the reviewers’ comments and our own corrections were highlighted in red in this revised version.

The manuscript “Oral administration of bovine lactoferrin modulates the effects of chronic stress on the lung immune response” by Mariazell Yépez-Ortega  et al. adds new information on immune reconstituting properties of lactoferrin. Although the work deals predominantly with the effects of Lf on the status of the innate immunity of lungs, modified by the immobilization stress, it also measures levels of serum Ig and corticosterone and body weight. So, the systemic parameters are also determined.

The language needs correction by a native speaker. There are numerous grammar and stylish errors e.g. “The analysis of CD 64/86 cell populations (Fig. 6B) diminished in RS, while it was increased in bLF+RS “.

We agree with the reviewer that language and style improvements are essential to enhance readability. To address this, the entire manuscript has been thoroughly revised and edited by the professional Language and Figure Editing service at MDPI Author Services.

The manuscript is difficult to read. The Introduction should be divided by several sections.

As suggested, we have divided the introduction into several thematic paragraphs to present the background information in a more logical and reader-friendly manner.

No sex of the mice is given. Males and females respond differently to stress (Yajun Qiao, et al. A Study of Sex Differences in the Biological Pathways of Stress Regulation in Mice CNS Neurosci Ther. 2025;31(5):e70433)

We thank the reviewer for this valuable observation. We added the sex of mice in the Material and Methods

Thank you for raising this important observation and for pointing out the recent work by Qiao et al. We have added the following to the Discussion:

“We employed the chronic restraint stress protocol in male mice to obtain more consistent results and accurately evaluate the effect of bLf on serum corticosterone levels. Many authors have reported discrepancies in the levels of stress mediators and cellular responses between male and female mice, and the evidence strongly suggests that the stress response can be modified by hormones associated with the estrous cycle”

REFERENCES

  • Qiao, Y.; Chen, H.; Guo, J.; Zhang, X.; Liang, X.; Wei, L.; Wang, Q.; Bi, H.; Gao, T. A Study of Sex Differences in the Biological Pathways of Stress Regulation in Mice. CNS Neurosci Ther 2025, 31, doi:10.1111/cns.70433.
  • Brivio, E.; Kos, A.; Ulivi, A.F.; Karamihalev, S.; Ressle, A.; Stoffel, R.; Hirsch, D.; Stelzer, G.; Schmidt, M. V.; Lopez, J.P.; et al. Sex Shapes Cell-Type-Specific Transcriptional Signatures of Stress Exposure in the Mouse Hypothalamus. Cell Rep 2023, 42, 112874, doi:10.1016/j.celrep.2023.112874.
  • Mitsushima, D.; Yamada, K.; Takase, K.; Funabashi, T.; Kimura, F. Sex Differences in the Basolateral Amygdala: The Extracellular Levels of Serotonin and Dopamine, and Their Responses to Restraint Stress in Rats. European Journal of Neuroscience 2006, 24, 3245–3254, doi:10.1111/j.1460-9568.2006.05214.x.
  • Heck, A.L.; Handa, R.J. Sex Differences in the Hypothalamic–Pituitary–Adrenal Axis’ Response to Stress: An Important Role for Gonadal Hormones. Neuropsychopharmacology 2019, 44, 45–58, doi:10.1038/s41386-018-0167-9

What do you mean by “oral deposition”. May be “buccal”?

We appreciate the reviewer’s attention to this wording. To avoid ambiguity, we have replaced the phrase with “buccal deposition” throughout the manuscript.

In the control group the mice should be also devoid of drinking and eating during the “stress periods”

We thank the reviewer for this valuable comment. In our experimental design, to ensure proper comparability, we have now clarified in the Materials and Methods section that the control group was also kept under the same conditions, without access to food and water during the Restraint Stress protocol.

More details is needed regarding the immobilization cages. 50 ml conical, perforated tubes?

We thank the reviewer for requesting this clarification. In our study, immobilization stress was applied using cylindrical plexiglass movement restriction chambers (9 cm large, 3 cm high and 3.5 cm diameter), with many holes for adequate ventilation, which restricted movement but did not caused pain or injury. This method is commonly used in stress protocols and has been validated in previous studies

  • Jarillo-Luna, A.; Rivera-Aguilar, V.; Garfias, H.R.; Lara-Padilla, E.;Kormanovsky, A.; Campos-Rodríguez, R. Effect of Repeated RestraintStress on the Levels of Intestinal IgA in Mice. Psychoneuroendocrinology 2007, 32, 681–692,doi:10.1016/j.psyneuen.2007.04.009.

Section 2.3. Fig 3B does not show results from TBL but from serum

R: The reviewer is right, we have corrected the text in Section 2.3 to describe the source of the data accurately, and we have also revised the figure legend to avoid any confusion.

The discussion does not mention potential advantageous effects of LF on the intestinal microbiota (Griffiths EA, et al. In vitro growth responses of bifidobacteria and enteropathogens to bovine and human lactoferrin. Dig Dis Sci 48: 1324-1332, 2003.; Kawasaki Y, et al.. Inhibitory effects of bovine lactoferrin on the adherence of enterotoxigenic Escherichia coli to host cells. Biosci Biotechnol Biochem 64: 348-354, 2000.) which may have a positive effect on the intestine microbiota during stress (Karla Vagnerová et al. Interactions Between Gut Microbiota and Acute Restraint Stress in Peripheral Structures of the Hypothalamic-Pituitary-Adrenal Axis and the Intestine of Male Mice. Front Immunol. 2019, 19:10:2655.) and consequently ameliorate negative effects of stress.

R: We thank the reviewer for providing us with your critical comments on this insightful suggestion and for highlighting these important references. We have expanded the Discussion to incorporate this point, and the papers were citated in our study:

bLf is also capable of modulating corticosterone release by regulating intestinal microbiota. Studies performed in germ-free mouse models have shown that microbiota downregulate serum corticosterone. Although the microbicidal effects of lactoferrin have been documented in in vitro and in vivo models, numerous reports highlight the prebiotic ability of bLf, which promotes the growth of Bacteroides and Parasutterella. These effects seem to be related to lactoferrin dose and iron saturation. In this regard, the effect of stress on the intestinal microbiota is not well understood, as differences in reports exist depending on the strain of mice, stress model, and duration of exposure. However, further studies are required to determine the exact mechanisms by which bLf modulates the microbiota and whether this is related to corticosterone release in this specific model of chronic stress”.

  • Griffiths, E.A.; Duffy, L.C.; Schanbacher, F.L.; Dryja, D.; Leavens, A.; Neiswander, R.L.; Qiao, H.; DiRienzo, D.; Ogra, P. In Vitro Growth Responses of Bifidobacteria and Enteropathogens to Bovine and Human Lactoferrin. Dig Dis Sci 2003, 48, 1324–1332, doi:10.1023/A:1024111310345.
  • Vagnerová, K.; Vodička, M.; Hermanová, P.; Ergang, P.; Šrůtková, D.; Klusoňová, P.; Balounová, K.; Hudcovic, T.; Pácha, J. Interactions Between Gut Microbiota and Acute Restraint Stress in Peripheral Structures of the Hypothalamic–Pituitary–Adrenal Axis and the Intestine of Male Mice. Front Immunol 2019, 10, doi:10.3389/fimmu.2019.02655.
  • Ergang, P.; Vagnerová, K.; Hermanová, P.; Vodička, M.; Jágr, M.; Šrůtková, D.; Dvořáček, V.; Hudcovic, T.; Pácha, J. The Gut Microbiota Affects Corticosterone Production in the Murine Small Intestine. Int J Mol Sci 2021, 22, 4229, doi:10.3390/ijms22084229.
  • Rabot, S.; Jaglin, M.; Daugé, V.; Naudon, L. Impact of the Gut Microbiota on the Neuroendocrine and Behavioural Responses to Stress in Rodents. OCL 2016, 23, D116, doi:10.1051/ocl/2015036.
  • Kawasaki, Y.; Tazume, S.; Shimizu, K.; Matsuzawa, H.; Dosako, S.; Isoda, H.; Tsukiji, M.; Fujimura, R.; Mu-ranaka, Y.; Ishida, H. Inhibitory Effects of Bovine Lactoferrin on the Adherence of Enterotoxigenic Escherichia Coli to Host Cells. Biosci Biotechnol Biochem 2000, 64, 348–354, doi:10.1271/bbb.64.348.
  • Molotla-Torres, D.E.; Hernández-Soto, L.M.; Guzmán-Mejía, F.; Godínez-Victoria, M.; Drago-Serrano, M.E.; Aguirre-Garrido, J.F. Oral Bovine Lactoferrin Modulation on Fecal Microbiota of Mice Underwent Immobilization Stress. J Funct Foods 2022, 95, 105153, doi:10.1016/j.jff.2022.105153.
  • Liu, Z.-S.; Chen, P.-W. Featured Prebiotic Agent: The Roles and Mechanisms of Direct and Indirect Prebiotic Activities of Lactoferrin and Its Application in Disease Control. Nutrients 2023, 15, 2759, doi:10.3390/nu15122759.
  • Watanabe, Y.; Arase, S.; Nagaoka, N.; Kawai, M.; Matsumoto, S. Chronic Psychological Stress Disrupted the Composition of the Murine Colonic Microbiota and Accelerated a Murine Model of Inflammatory Bowel Disease. PLoS One 2016, 11, e0150559, doi:10.1371/journal.pone.0150559.

Round 2

Reviewer 1 Report

Comments and Suggestions for Authors

All issues raised have been addressed.